# Generative Hierarchical Materials Search

**Sherry Yang,**[*] **Simon Batzner, Ruiqi Gao, Muratahan Aykol, Alexander Gaunt,**
**Brendan McMorrow**, **Danilo Rezende**, **Dale Schuurmans**, **Igor Mordatch**, **Ekin Dogus Cubuk**
Google DeepMind

## Abstract

Generative models trained at scale can now produce text, video, and more recently, scientific data such as crystal structures. In applications of generative approaches to materials science, and in particular to crystal structures, the guidance from the domain expert in the form of high-level instructions can be essential for an automated system to output candidate crystals that are viable for downstream research. In this work, we formulate end-to-end language-to-structure generation as a multi-objective optimization problem, and propose Generative Hierarchical Materials Search (GenMS) for controllable generation of crystal structures. GenMS consists of (1) a language model that takes high-level natural language as input and generates intermediate textual information about a crystal (e.g., chemical formulae), and (2) a diffusion model that takes intermediate information as input and generates low-level continuous value crystal structures. GenMS additionally uses a graph neural network to predict properties (e.g., formation energy) from the generated crystal structures. During inference, GenMS leverages all three components to conduct a forward tree search over the space of possible structures. Experiments show that GenMS outperforms other alternatives of directly using language models to generate structures both in satisfying user request and in generating low-energy structures. We confirm that GenMS is able to generate common crystal structures such as double perovskites, or spinels, solely from natural language input, and hence can form the foundation for more complex structure generation in near future.

## 1 Introduction

Modern technologies increasingly rely on the development of materials, such as semiconductors [1], solar cells [2], and lithium batteries [3]. Large-scale generative models, trained on expansive internet data, exhibit intriguing generalization capabilities. For example, these models can synthesize a highly realistic image of "an astronaut riding a horse" by merging two distant concepts [4]. This raises a compelling question: can the generalization capabilities of large generative models, pretrained on existing materials science knowledge, be harnessed to combine knowledge from existing materials systems to propose candidate crystals?

Previous research has demonstrated that generative models can output crystal structures that are not in the the training data [5, 6, 7]. However, these works typically require either a vast number of unconditional samples to generate an unknown material [5, 8] or a chemical formula provided during inference [6, 9]. It is difficult for end users to come up with new chemical formulae, as it is hard to know which compositions will result in what material properties. Therefore, it is highly desirable to develop an interface that allows users to describe the desired characteristics of crystal structures — such as properties, compositions, space groups, and geometric characteristics — in natural language. For example, a user might specify "a stable chalcogenide with atom ratio 1:1:2 that is not on ICSD." Ideally, a model should automatically interpret these high-level language instructions to search for,

---

[*]Correspondence to `sherryy@google.com` and `cubuk@google.com`.
38th Conference on Neural Information Processing Systems (NeurIPS 2024).

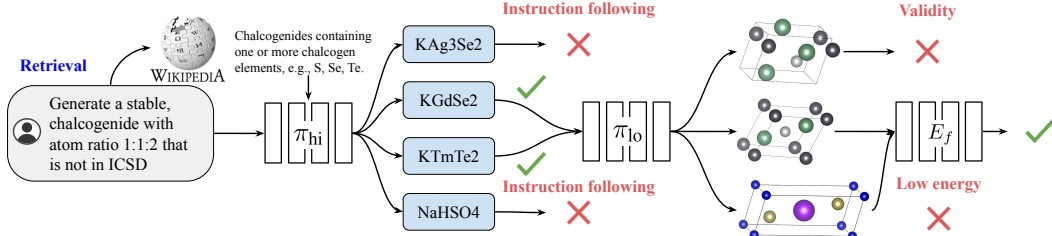

Figure 1: **Overview of GenMS.** GenMS takes a high-level language instruction as input, retrieves relevant information from the internet, and samples from a high-level LLM ($\pi_{hi}$) to generate candidate formulae that satisfy user requirement. GenMS then samples from a low-level diffusion model ($\pi_{lo}$) to generate structures conditioned on candidate formulae. Sampled structures then go through a property prediction module for selection.

generate, and validate a wide range of potential structures, ultimately producing one that best meets the user's specifications.

However, developing an end-to-end language-to-structure generative model presents several challenges, for which we make a few key observations. First, there are no existing labeled datasets that map language descriptions directly to crystal structures. Nevertheless, we observe that there is a wealth of language-to-formula data available online, including Wikipedia articles, research papers, and textbooks. This data can be complemented by formula-to-structure information from specialized materials databases such as the Materials Project [10], ICSD [11], OQMD [12], etc. Second, the task of converting language into structures is inherently multimodal, requiring the transformation of discrete linguistic inputs to continuous structural outputs. Nevertheless, it has been shown that semantic-level autoregressive models combined with low-level (pixel-level) diffusion models are effective for cross-modal generation, such as in text-to-video applications [13, 14]. Lastly, user descriptions of desired crystal structures can often be vague — users may not articulate all relevant details about the crystal they wish to generate. We observe that one can leverage generative models to infer missing information, and rely on additional search and selection mechanisms to identify structures that best satisfy a user's requirement.

Based on these observations, we propose Generative Hierarchical Materials Search (GenMS) for end-to-end language-to-structure generation. GenMS consists of (1) a large language model (LLM) pretrained on high-level materials science knowledge from the internet, (2) a diffusion model trained on low-level crystal structure databases, and (3) a graph neural network (GNN) for property prediction. To improve the efficiency of (2), GenMS proposes a compact representation of crystal structures for diffusion models. During inference, GenMS prompts the LLM to generate candidate chemical formulae according to user specification, samples structures from the diffusion model, and uses the GNN to predict the properties of the sampled structures. To sample structures that best satisfy user requirements during inference, we formulate language-to-structure as a multi-objective optimization problem, where user specifications are transformed into objectives that can be optimized at both the formula and structure level.

We first evaluate GenMS's ability to generate crystal structures from language instructions, and find that GenMS can successfully generate structures that satisfy user requests more than 80% of the time for three major families of structures, while proposing structures with low formation energies, as verified by DFT calculations. In contrast, using pretrained LLMs to directly generate crystal structures from user instructions in a zero-shot manner often results in close to a 0% success rate. Qualitative evaluations show that GenMS is able to generate complex structures, such as layered structures, double perovskites, and spinels, solely from natural language. We next study the effect of each individual component of GenMS. Here we find that language instructions have a significant impact on the structures generated, that the novel compact representation of crystals proposed by GenMS improves the DFT convergence rate of diffusion generated crystal structures by 50% over previous work, and that using a pretrained GNN to select samples leads to lower energy structures more than 80% of the time. Given such experimental evidence, we believe the development of language-to-structure models are promising for enabling users to find viable crystal structure candidates, complementing existing databases in utility

# 2 Generative Hierarchical Materials Search

We begin by formulating the problem of generating crystal structures from high-level language as a multi-objective optimization task. Given this formulation, we then propose a hierarchical, multi-modal tree search algorithm that leverages language models, diffusion models, and graph neural networks as submodules. Lastly, we discuss the specific design choices for each of the submodules.

## 2.1 Language to structure as a multi-objective optimization

Given some high-level language description $g \in \mathcal{G}$ of desired structures, we want to learn a conditional crystal structure generator $\pi(\cdot|g) : \mathcal{G} \mapsto \Delta(\mathcal{X})^2$ that can be used to sample crystal structures $x \in \mathcal{X}$ conditioned on language. One option is to parametrize $\pi$ with a pretrained LLM. However, pretrained LLMs alone are not able to predict sufficiently accurate crystal structures, due to the lack of low-level structural information about crystals (e.g., 3D atom coordinates) in the pretraining data.

If we had access to a paired language-to-structure dataset, $\mathcal{D} = \{g_i, x_i\}_{i=1}^N$, $\pi$ could be trained using a maximum likelihood objective. However, materials data naturally exist at different levels of abstraction and are segregated into different sources: high-level symbolic knowledge is documented in sources like Wikipedia articles, research papers, and textbooks, whereas detailed low-level crystal information, including continuous-valued atom positions, is stored in specialized crystal databases like the Materials Project [10] and ICSD [11]. Even though a direct language-to-structure dataset $\mathcal{D}$ remains unavailable, the pretraining data for LLMs, including Wikipedia articles, research papers, and textbooks, can be viewed as a high-level symbolic dataset $\mathcal{D}_{\mathrm{hi}} = \{g_i, z_i\}_{i=1}^m$, where $z \in \mathcal{Z}$ denotes symbolic textual information such as chemical formulae. Meanwhile, many crystal databases already feature paired data, $\mathcal{D}_{\mathrm{lo}} = \{z_i, x_i\}_{i=1}^n$, linking chemical formulae to detailed crystal structures.

Given this observation, we propose to factorize the crystal generator as $\pi = \pi_{\mathrm{hi}} \circ \pi_{\mathrm{lo}}$, where $\pi_{\mathrm{hi}} : \mathcal{G} \mapsto \Delta(\mathcal{Z})$ and $\pi_{\mathrm{lo}} : \mathcal{Z} \mapsto \Delta(\mathcal{X})$, so that $\pi_{\mathrm{hi}}$ and $\pi_{\mathrm{lo}}$ can be trained using different datasets $\mathcal{D}_{\mathrm{hi}}$ and $\mathcal{D}_{\mathrm{lo}}$. Furthermore, we consider two heuristic functions, $R_{\mathrm{hi}}(g, z) : \mathcal{G} \times \mathcal{Z} \mapsto \mathbb{R}$ and $R_{\mathrm{lo}}(z, x) : \mathcal{Z} \times \mathcal{X} \mapsto \mathbb{R}$, where the high-level heuristic function $R_{\mathrm{hi}}$ can be used to select formulae that satisfy the language input at a high level, and the low-level heuristic function $R_{\mathrm{lo}}$ can be used to select structures that are both valid and exhibit desirable properties such as low formation energy. To this end, we propose to search for crystal structure given language input by finding a chemical formula / space group $z$ with a corresponding crystal structure $x$ that jointly optimize

$$z^*, x^* = \arg \max_{z, x \sim \pi_{\mathrm{hi}}, \pi_{\mathrm{lo}}} \mathbb{E}_{z \sim \pi_{\mathrm{hi}}, x \sim \pi_{\mathrm{lo}}(z)}[\lambda_{\mathrm{hi}} \cdot R_{\mathrm{hi}}(g, z) + \lambda_{\mathrm{lo}} \cdot R_{\mathrm{lo}}(z, x)], \tag{1}$$

where $\lambda_{\mathrm{hi}}$ and $\lambda_{\mathrm{lo}}$ are hyperparameters to control how much weight to put on high and low-level heuristics. Note that $R_{\mathrm{hi}}$ and $R_{\mathrm{lo}}$ can also be combinations of multiple objectives. For instance, $R_{\mathrm{hi}}$ can be a weighted sum of instruction following and simplicity, where $R_{\mathrm{lo}}$ can be a weighted sum of properties such as band gap, conductivity, and formation energy.

## 2.2 Searching through language and structure

Given the objective in Equation 1, it is clear that a pretrained LLM (even with finetuning) is insufficient to optimize for the best structure $x^*$. Instead, we propose to first sample a set of intermediate chemical formulae from a pretrained LLM $\pi_{\mathrm{hi}}(g)$ conditioned on language input $g$. We then use the high-level heuristic function $R_{\mathrm{hi}}$ to prune and rank the intermediate formulae. In practice, $R_{\mathrm{hi}}$ is a combination of (i) a regular expression checker (to ensure sampled formulae are valid chemical formulae), (ii) a uniqueness checker against formulae from existing crystal datasets such as Materials Project and ICSD, and (iii) a formula compliance checker to ensure the sampled formulae are compatible with user request (e.g., atom ratio 113 for perovskites, 227 for pyrochlore, and 124 for spinel). For formulae that pass these checks, we prompt a pretrained LLM as $R_{\mathrm{hi}}$ to rank the formulae by how likely they are to comply with the user request $g$. We then select the top $W$ ranked formulae to generate $L$ crystal structures each using $\pi_{\mathrm{lo}}$ parametrized by a diffusion model, and use a graph neural network $R_{\mathrm{lo}}$ to rank the $W \times L$ structures by their predicted formation energy. Note that additional checkers can be integrated in $R_{\mathrm{lo}}$, such as structural and compositional validity defined in [5]. We illustrate the overall search procedure in Algorithm 1.

---

²We use $\Delta(\cdot)$ to denote a probability simplex function.

---

**Algorithm 1** Generative Hierarchical Materials Search

---

1: **Input:** Language input $g$
2: **Functions:** High-level language policy $\pi_{\mathrm{hi}}(z|g)$, high-level heuristic function $R_{\mathrm{hi}}(g, z)$, low-level diffusion policy $\pi_{\mathrm{lo}}(x|z)$, low-level heuristic function $R_{\mathrm{lo}}(z, x)$.
3: **Hyperparameters:** High-level language branching factor $H$, low-level structure branching factor $L$, max width for formulae $W$.
4: plans $\leftarrow [[g] \ \forall \ i \in \{1 \ldots H\}]$      # Initialize H different plans starting with language input.
5: **for** $h = 1 \ldots H$ **do**
6:    $g \leftarrow$ plans$[h][-1]$      # Get the high-level language specification from the tree.
7:    $\{z_i\}_{i=1}^{H} \leftarrow \pi_{\mathrm{hi}}(g)$      # Generate $H$ different intermediate formulae.
8:    $z^* = \mathrm{argmax}(\{g, z_i\}_{i=1}^{H}, R_{\mathrm{hi}})$
9:    plans$[h]$.append$(z^*)$      # Add formula with the best heuristic value to plan.
10: **end for**
11: plans $\leftarrow$ sort(plans, $R_{\mathrm{hi}}$)      # Sort formulae based on heuristic.
12: **for** $w = 1 \ldots W$ **do**
13:    $z \leftarrow$ plans$[w][-1]$      # Get the best intermediate formula from the tree.
14:    $\{x_i\}_{i=1}^{L} \leftarrow \pi_{\mathrm{lo}}(z)$      # Generate $L$ low-level structures.
15:    $x^* = \mathrm{argmax}(\{z, x_i\}_{i=1}^{H}, R_{\mathrm{lo}})$
16:    plans$[w]$.append$(x^*)$      # Add structure with the best heuristic value to plan.
17: **end for**
18: **return** plans$[0][0]$      # Return the best structure.

---

**Alternative search strategies.** The search algorithm described above, Algorithm 1, follows the best-first search strategy, i.e., intermediate formulae and final structures are sorted and searched over based on the preference of a heuristic function. Alternative search strategies such as breadth-first or depth-first can also be employed. The most suitable search strategy depends on the downstream application and computational resources available. For instance, if large-scale density function theory (DFT) calculations are available downstream, we can employ breadth-first search to devise more diverse composition.

**Prevent heuristic exploitation.** One concern of using a heuristic GNN to select structures with the lowest formation energy is that the GNN might exploit irregularities in the predicted structures, especially when a predicted structure lies outside of the training manifold of the energy GNN. To mitigate this issue, we use the GNN pretrained by [15] on DFT energies and forces of unrelaxed structures (hence the GNN has seen more irregular structures prior to relaxation.) Furthermore, we discard sampled structures from $\pi_{\mathrm{lo}}$ if they result in energy predictions from $R_{\mathrm{lo}}$ that lie outside of a threshold range.

## 2.3 Choices of parametrization for the submodules

Since controllable crystal structure generation from language input is multimodal by nature, there are various design choices for the parametrization of the submodules in Equation 1, namely the generators $\pi_{\mathrm{hi}}, \pi_{\mathrm{lo}}$ and the heuristic functions $R_{\mathrm{hi}}, R_{\mathrm{lo}}$. In this section, we discuss the parametrization choices we have found to be the most effective.

**Retrieval augmentation and long-context deduplication.** One important recent advance in LLMs is increased context length [16]. The factorization $\pi = \pi_{\mathrm{hi}} \circ \pi_{\mathrm{lo}}$ provides a natural way to integrate additional context in $\pi_{\mathrm{hi}}$ via long-conext generation. Specifically, we further factorize $\pi_{\mathrm{hi}}$ into $\pi_{\mathrm{hi}} = \pi_{\mathrm{hi}}^{\mathrm{retrival}} \circ \pi_{\mathrm{hi}}^{\mathrm{RAG}}$, where $\pi_{\mathrm{hi}}^{\mathrm{retrival}}(\cdot|g)$ is a deterministic retrieval function that uses the Wikipedia API to retrieve textual information related to language input $g$, while $\pi_{\mathrm{hi}}^{\mathrm{RAG}}$ is a retrieval augmented generative (RAG) model that proposes chemical formulae and space groups conditioned on the information retrieved from the internet. Another use case for long-context LLMs is to further encourage the generation of *new* compositions by providing the formulae for all known crystals in the context, then asking $\pi_{\mathrm{hi}}$ to produce a formula that is not in the context. As we will see in Section 3.2, this drastically improves the efficiency of the search, as a large subset of the search space with known crystals can be eliminated.

**Compact crystal representation.** In order to support efficient tree search at inference time, we need to ensure that sampling from both $\pi_{\mathrm{hi}}$ and $\pi_{\mathrm{lo}}$ are efficient. Previous work on diffusion models

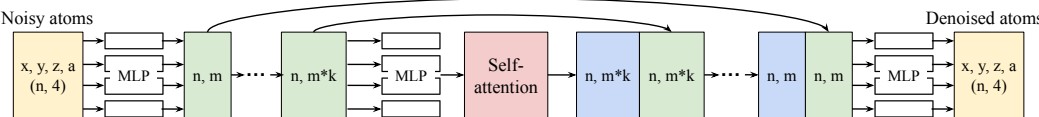

Figure 2: **Diffusion architecture with compact crystal representation.** The diffusion model in GenMS represents crystal structures by the $x, y, z$ location of each atom plus the atom number $a$ represented as a continuous value. Each atom undergoes blocks consisting of multi-layer perceptrons followed by order-invariant self-attention. The MLP and self-attention blocks are repeated $k$ times where each repetition increases the dimension of the hidden units. The concatenation of skip connections are employed as in other U-Net architectures.

for crystal structure generation has leveraged sparse data structures, such as voxel images [17, 18, 19], graphs [5], and periodic table shaped tensors [6]. These existing representations of crystals incur computational overhead due to sparsity (voxel images, padded tensors) or quadratic complexity as the number of atoms in the system increases (graphs). Instead, we propose a new compact representation of crystal structures, where each crystal $x \in \mathcal{X} \subset \mathbb{R}^{A \times 4}$ is represented by a $A \times 4$ tensor, with $A$ being the number of atoms in the crystal, and the inner 4 dimensions representing the $x, y, z$ location of an atom along with its atom number. Here we directly represent the atom number as a continuous value normalized to the range of the input in the diffusion model to further improve inference speed, as opposed to representing the atom number using a one-hot vector. In addition, we use another $2 \times 3$ vector to represent the lattice structure (i.e., angles and lengths of the unit cell). Figure 2 illustrates the architecture for the diffusion model with compact crystal representations, where each atom undergoes multi-layer perceptron (MLP) followed by order-invariant self-attention (without positional encoding) across atoms. Different from typical U-Net architecture for image generation, there is no downsampling or upsampling passes that change the input resolution. Nevertheless, we follow the concatenation of skip connections commonly used in U-Net architectures [20]. Additional details and hyperparameters for the diffusion model can be found in Appendix A.3.

## 3 Experimental Evaluation

We now evaluate the ability of GenMS to generate crystal structures from high-level language descriptions. First, we evaluate the success of end-to-end generation in Section 3.1. We then investigate the individual components of GenMS in Section 3.2. See details of experimental setups in Appendix A.

### 3.1 End-to-end evaluation

**Baselines and metrics.** We aim to evaluate GenMS's ability to generate unique, valid, and potentially stable crystal structures from well-known crystal families that satisfy high-level language specifications. We consider few-shot prompting of LLMs to generate crystal information files (CIF) as a baseline. Specifically, we give the Gemini long context model [16] a number of CIF files from a particular crystal family, as specified by language input as prompt, with the number of CIF files ranging from 1, 5, 25 to as many as can fit in the context. We ask the LLM to generate 100 samples given each language instruction. See additional details of baselines in Appendix A.2. We do not compare to finetuning LLMs to generate CIF files in this section, as there are no high-level language to low-level crystal structure datasets available for finetuning such an instruction following LLM. Nevertheless, we will compare the diffusion model in GenMS to formula-conditioned structure generation using finetuned LLM in Section 3.2. We consider language input that directs the model to generate unique and stable crystals from a particular crystal family (perovskite, pyrochlore, and spinel). We consider the following metrics for evaluation: (i) CIF validity, which measures whether the generated CIF file can be properly parsed by pymatgen parser [21]. (ii) Structural and composition validity, which verify atom distances and charge balances using SMACT [22], following [5]. (iii) Formation energy per atom ($E_f$) in the unit of eV/atom, which measures the stability of predicted structures using a pretrained GNN. We further conduct DFT calculations to compute $E_f$ (see details in Appendix A.4) for structures predicted by GenMS. (iv) Uniqueness, which measures the percentage of generated formulae that do not exist in Materials Project [10] or ICSD [11]. Finally, (v) the match rate, which measures the percentage of generated structures that can be matched (according to the pymatgen

| Family | Metric | Prompting CIF | | | | GenMS |
|---|---|---|---|---|---|---|
| | | 1 shot | 5 shot | 25 shot | Max | |
| Perovskites | CIF validity ↑ | 0.94 | 1.00 | 0.98 | 0.88 | **1.00** |
| | Structural validity ↑ | 0.04 | 0.28 | 0.66 | 0.22 | **1.00** |
| | Composition validity ↑ | 0.07 | 0.17 | 0.45 | 0.00 | **0.85** |
| | $E_f$ (GNN/DFT) ↓ | N/A | -0.19 | 0.28 | 0.53 | **-0.47/-1.32** |
| | Uniqueness ↑ | 0.07 | 0.29 | 0.68 | 0.16 | **0.90** |
| | Match rate ↑ | 0.00 | 0.09 | 0.36 | 0.19 | **0.93** |
| Pyrochlore | CIF validity↑ | 0.40 | 0.60 | 0.64 | 0.88 | **1.00** |
| | Structural validity↑ | 0.36 | 0.36 | 0.28 | 0.23 | **0.95** |
| | Composition validity↑ | 0.00 | 0.22 | 0.18 | 0.00 | **0.89** |
| | $E_f$ (GNN/DFT)↓ | 1.28 | 1.19 | 0.63 | -1.22 | **-1.37/-2.56** |
| | Uniqueness↑ | 0.36 | 0.38 | 0.45 | 0.08 | **0.49** |
| | Match rate ↑ | 0.00 | 0.00 | 0.04 | 0.00 | **0.86** |
| Spinel | CIF validity | 0.73 | 0.96 | 0.97 | 0.96 | **1.00** |
| | Structural validity↑ | 0.47 | 0.61 | 0.71 | 1.00 | **1.00** |
| | Composition validity↑ | 0.29 | 0.92 | 0.92 | 1.00 | **1.00** |
| | $E_f$ (GNN/DFT)↓ | 1.09 | -0.85 | -0.97 | -1.37 | **-1.38/-1.77** |
| | Uniqueness↑ | 0.48 | 0.13 | **0.51** | 0.08 | 0.44 |
| | Match rate ↑ | 0.00 | 0.12 | 0.18 | 0.08 | **0.89** |

Table 1: **End-to-end evaluation** of generating crystal structure from natural language. GenMS significantly outperforms LLM prompting baselines in producing unique and low-energy (predicted by GNN) structures that satisfy user request. We further conduct DFT calculation to compute $E_f$ (formation energy in eV/atom) averaged across structures generated by GenMS. Values before "/" in the row "(GNN/DFT)" represent GNN predicted $E_f$, and after "/" represent DFT computed $E_f$. We report $E_f$ from GNN prior to relaxation, and $E_f$ from DFT post relaxation. DFT calculations for baselines are eliminated as many structures from the baselines do not follow user instruction. N/A represents $E_f$ predicted by GNN falling outside of the reasonable range.

structure matcher) to one of the structures of the corresponding family in Materials Project. More details of these metrics can be found in Appendix A.1.

**Results on specifying crystal family.** The evaluation of GenMS and baselines are shown in Table 1. Since GenMS does not rely on an LLM to directly generate CIF files, the compact crystal representation (described in Section 2.3) always results in structures that can be parsed by pymatgen (100% CIF validity). In addition, structures generated by GenMS have a much higher validity and match rate compared to those generated by the baselines. GenMS struggles slightly with uniqueness, as less than half of the generated formulae for pyrochlore and spinel are unique with respect to MP and ICSD. Structures produced by GenMS have lower average $E_f$. Increasing the number of CIF files in the context generally improves the performance of the baselines (1, 5, and 25-shot), but including too many files in the context can hurt performance (Prompting CIF Max).

**Qualitative evaluation.** In addition to the three families of structures evaluated above, we qualitatively evaluated GenMS's ability to generate structures that satisfy ad hoc user requests, such as "a layered material", "an elpasolite", and so on. GenMS can consistently produce structures that satisfy user request as shown in Figure 3, and have plausible initial geometries. Interestingly, we observe that GenMS can understand semantic-level request, suggesting more "fluoride" like chemistries when asked for "elpasolite", which is reasonable as elpasolite is associated with the mineral K2NaAlF6.

**Effect of search.** Next, we aimed to understand the effect of search in GenMS, especially in producing low-energy structures. For each of the family of crystals in Table 1, we analyzed the effect of the language and structure branching factors (H and L in Algorithm 1). Only crystals that match input specification were considered for energy computation. We found that increasing the branching factor of both language and structure enables GenMS to generate structures with lower formation energies (at a higher inference cost).

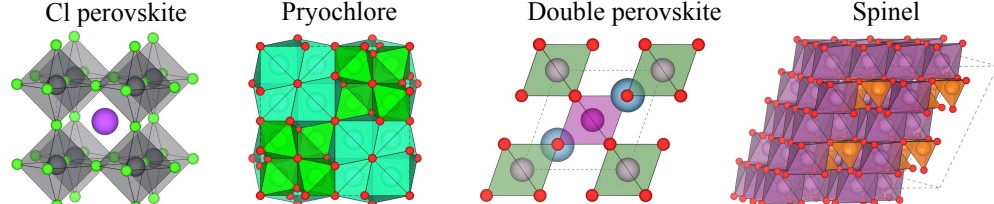

Cl perovskite          Pryochlore          Double perovskite          Spinel

Figure 3: **Qualitative evaluation.** We test GenMS on a set of ad hoc language inputs to generate plausible examples from well-known crystal families. GenMS is able to search for the corresponding structures that satisfy user requests and have plausible initial geometries. Visualization provided by VESTA [23].

| | Perovskite | | | Pyrochlore | | | Spinel | |
|---|---|---|---|---|---|---|---|---|
| Language branch (H) | Structure branch (L) | $E_f$ (DFT) | Language branch (H) | Structure branch (L) | $E_f$ (DFT) | Language branch (H) | Structure branch (L) | $E_f$ (DFT) |
| 1 | 1 | -0.55 | 1 | 1 | -2.40 | 1 | 1 | -1.67 |
| 1 | 100 | -0.79 | 1 | 100 | -2.51 | 1 | 100 | -1.74 |
| 25 | 1 | -2.76 | 25 | 1 | -3.02 | 25 | 1 | -1.82 |
| 25 | 100 | **-2.91** | 25 | 100 | **-3.24** | 25 | 100 | **-1.95** |

Table 2: $E_f$ **(computed by DFT) vs. branching factor.** GenMS can generate structures with lower formation energy (computed by DFT) at the cost of slower inference when language and structure branching factors are increased.

## 3.2 Evaluating individual components of GenMS

Next, we evaluate the individual component of GenMS, including the effect of using language to narrow down the search space, the choice of the compact representation of crystal structures, and finally the best-of-N sampling strategy for choosing the crystal structures with low formation energy.

**Effect of language.** We want to understand whether GenMS can provide effective control over formulae proposed by the LLM at the semantic-level through natural language. In Table 3, we first show that requesting a particular element to be in the formula always results in formulas with that particular element being proposed by the pretrained LLM $\pi_{hi}$. We then show that when a user requests for metal, the model is 4 times more likely to generate formulae for metal. The model also respects a user's request for the generated formulae to be unique (with respect to either a user provided list of known formulae in the context of the LLM, or the name of some crystal database).

Next, we study the effect of retrieval augmented generation (RAG). We use GenMS with and without RAG to propose 25 formulae for each of the three major crystal families from Section 3.1 and generates 4 structures per formula using the diffusion model. We report the rate of valid formulae proposed by the LLM and the structures that can be matched with existing structures from the corresponding family in Table 4. RAG improves both the rate of valid formulae and matched structures.

| | Element constraint | Metal only | Unique (custom list) | Unique (Materials Project) |
|---|---|---|---|---|
| Not asking | N/A | 0.25 | 0.24 | 0.16 |
| Asking | **1.00** | **1.00** | **0.88** | **0.96** |

Table 3: **Effect of language.** Asking for a specific element from the periodic table results in formulae that always contain that element. Asking for metal and formulae unique with respect to some existing formula sets result in formulae that are more likely to satisfy user requests.

| | Valid formula | Match rate |
|---|---|---|
| Without RAG | 0.97 | 0.72 |
| With RAG | **1.00** | **0.89** |

Table 4: **Effect of RAG.** Using retrieval augmented generation improves the percentage of valid formulae and matched structures. See details for the structure matcher used in Appendix A.1.

|            | UniMat [6]       | GenMS               |
| ---------- | ---------------- | ------------------- |
| DFT converge | 0.62           | **0.93**            |
| Mean $E_f$  | $-0.40 \pm 0.06$ | $\mathbf{-0.49 \pm 0.03}$ |

|               | Small LLM [9] | Large LLM | GenMS    |
| ------------- | ------------- | --------- | -------- |
| Success       | 0.86          | 0.87      | **0.93** |
| Match (unseen) | 0.26         | 0.37      | **0.48** |

Table 5: **DFT evaluation of GenMS vs Uni-Mat.** Structures proposed by GenMS result in much high DFT convergence rate and lower average $E_f$ than structures proposed by Uni-Mat. Error bars reflect standard error.

Table 6: **Comparison to finetuned LLMs.** GenMS's diffusion model with compact representations achieves high success rate of generating valid crystals, as well as a high matching rate to holdout structures compared to CrystalLM [9].

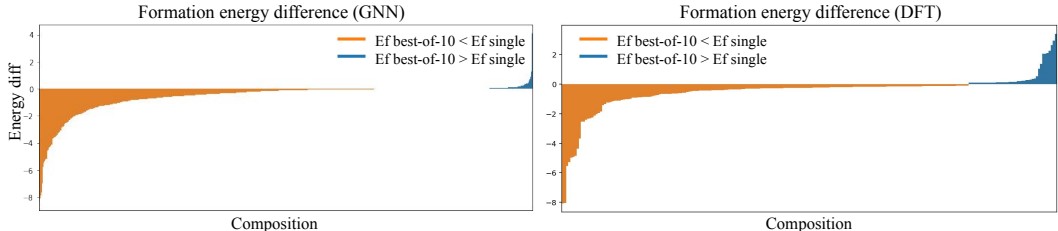

Figure 4: **Formation energy between Best-of-N and a single sample.** Both according to energy predicted by GNN and calculated by DFT, best-of-N with N = 10 leads to improvements in energy compared to single samples for 80% of 1,000 compositions considered.

**Compact crystal representation.** We now evaluate the diffusion model $\pi_{\text{lo}}$ trained using the compact representation of crystals structures described in Section 2.3. We compare diffusion model with compact crystal representation against two prior work for generating crystal structures conditioned on composition. UniMat [6] proposed a periodic table representation of crystals which requires a large amount of paddings to handle atoms that do not exist in the structure. CrystalLM [9] proposes to finetune an LLM to directly generate CIF files from input compositions. In Table 5, we report the DFT convergence rate and DFT calculated $E_f$ on a set of holdout structures following [6]. We observe that the compact crystal representation results in both higher convergence rate and lower $E_f$ than the sparse representation in [6]. To compare GenMS's diffusion model against finetuning LLMs to generate CIF files directly, we follow the experimental setting of CrystalLLM where we train a composition conditioned diffusion model on a combination of Materials Project [10], OQMD [24], and NOMAD [25], and test the success rate of generating matching structures for unseen compositions following [9]. In Table 6, we see that GenMS has significantly higher rate in producing a valid crystal and a crystal that can be matched to the test set in [9].

**Best-of-N structure sampling.** To better understand the effect of high structure branching factor in Algorithm 1 across different compositions, we measure the difference in the formation energy, using a holdout test set of 1,000 compositions, between using the energy prediction GNN to select the best of 10 samples compared to only predicting a single structure. The energy difference with and without best-of-N sampling is shown in Figure 4. Using best-of-N with $N = 10$ results in improved energy for over 80% of structures (as also verified by DFT calculations). We found the energy prediction GNN to be a good indicator of the true energy of the crystal structures, i.e., the GNN predicted energy difference (left) and the DFT calculated energy difference (right) are very similar in Figure 4.

## 4 Related work

**Hierarchical and latent image and video generation.** Image and video generative models have exhibited an impressive ability to synthesize photorealistic images or videos when given text description as input. Many of the state-of-the-art models adopt a hierarchical modeling approach that inspired the design of with GenMS. For example, latent diffusion models [26, 27] contains (1) a language model that converts text to high-level text embeddings, (2) a diffusion model takes the text embeddings as input and output latents in a compressed latent space, and (3) a feed forward decoder network [26] or a diffusion decoder [28, 14] that given the generated latents generates full-resolution signals in the pixel space. Cascaded diffusion models [29, 30, 31] instead proposed to generate signals at the lowest resolution with a standard diffusion model, followed by a few super-resolution

models that successively upsample signals and add high-resolution details. Similar to GenMS, by breaking down complicated image or video generation into a hierarchy of less challenging problems, these models can generate high quality samples more efficiently and effectively.

**Generative models for crystal structures.**  A number of works [9, 8, 32] have proposed to train or fine-tune language models to generate output files containing crystal information or low-level atom positions. However, it remains expensive and challenging to train and generate detailed structural information with LLMs. On the other hand, diffusion models, as a powerful class of generative model in vision, have been applied to generate crystal structures [5, 7, 6]. However these methods either reply on training with a large set of unconditional samples and brute-force sampling for new materials not in the training set, or necessitate predetermined compositions as conditioning information during inference. Handling of candidate structure generation requires a model capable of independent reasoning about chemical compositions based on high-level user specifications and structure optimization, as done in GenMS.

**Hierarchical search and planning.**  The problem of learning to generate low-level continuous output from high-level language instructions, while employing intermediate search and planning steps, has been studied in other domains such as continuous control [33], self-driving [34], and robotics [35]. While some works have focused on purely using LLMs to search and plan through complex output spaces [36, 37], other research has shown that solely relying on LLMs to search and plan can fail short due to the lack of low-level information (e.g., locations, precise motions) captured in the model [38]. Recently, video generation models have been applied to provide additional details about the physical world so that low-level control actions can be extracted more accurately [39, 40, 41, 42]. GenMS follows a similar approach but focuses on generating crstyal structures, using diffusion models on top of LLMs to provide additional details about crystal structure, enabling high-level plans (i.e., symbolic chemical formulae) to be verified at a low-level (i.e., crystal structures with precise atom locations).

**Large language models for science.**  Recently, there has been a surge of interest in applying large langauge models in domains of science, such as physics [43], biology [44], chemistry [45, 46], and materials science [47]. In these settings, LLMs generally serve as a conversational [44] or educational [48] tool, where LLMs output natural language to be consumed by human users (e.g., an answer to a scientic question asking about the property of some existing crystal structure). On the other hand, we are interested in the ability of a pretrained LLM to propose intermediate textual information such as chemical formulae for interesting crystal structures. Closest to our work are [49, 50] which leverage an LLM to generate SMILES or other chemical strings for molecular design. Nevertheless, we are interested in generating not just the formulae, but the actual crystal structures with continuous-valued atom locations, as many materials property can only be calculated and verified once the full structure available.

## 5 Conclusion and future work

We have introduced GenMS, an initial attempt at enabling end-to-end generation of candidate crystal structures that look physically viable and satisfy instructions expressed in natural language. GenMS can generate examples from families such as pyrochlores and spinels purely from natural language prompts. We hope the design principles of GenMS will initiate broad interest in exploiting language as a natural interface for flexible design and generation of crystal structures that meet user-specified criteria, and enable the domain experts to work more efficiently. GenMS has a few limitations that call for future work:

- **Generating complex structures.** While GenMS is able to generate simple structures such as those shown in Figure 3, we found that GenMS is less effective in generating complex structures such as Mxenes and Kagome lattices. Controllable generation of highly complex crystal structures is an interesting area of future work.

- **Impact on experimental exploration.** While we have shown that GenMS is effective in generating crystal structures that are not in public databases and that satisfy user requirements, its effectiveness in suggesting specific materials with target properties (e.g., battery electrodes or electrolytes, semiconductors, superconductors etc.) requires further experimental verification.

- **Synthesizability.** While the goal of GenMS is to provide an end-to-end generative framework from natural language instructions to realistic crystal structures, synthesizability of the generated crystals is not currently part of the pipeline. We foresee development in multimodal models and integration of other computational tools from materials science to allow predicted structures to be assessed for synthesizability.

- **Extension to other chemical systems.** We have shown that GenMS can effectively generate crystal structures from natural language. We note that GenMS can also potentially be extended to generating molecules and protein structures from natural language (e.g. "generate a protein with an alpha-helix"). We leave these explorations for future work.

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

# Appendix

## A  Experiment details

In this section, we provide additional experimental details, including metrics used for evaluation, baselines, architecture and training of the diffusion model with the compact crystal representation, and details of the setup for the DFT calculations.

### A.1  Details of evaluation metrics

**Structure and composition validity.**  The structure and composition validity metrics follow [5]. The structure validity determins that a structure is valid as long as the shortest distance between any pair of atoms is larger than 0.5 Å [19]. The composition is valid if the overall charge is neutral as computed by SMACT [22].

**Uniqueness.**  We determine a generated formula is unique if the reduced form of the formula does not exist in either Materials Project [10] or ICSD [11]. For instance, if ICSD contains formula in the form of AB2, we consider A2B4 generated by the model as a duplicate (thus not unique) structure.

**Match rate.**  To compute the match rate, we use the `StructureMatcher` module from pymatgen's analysis package. We set the hyperparameters of the matcher following [9], specifically with $\texttt{stol} = 0.5, \texttt{ltol} = 0.3, \texttt{angle\_tol} = 10$. For each family of crystals in perovskite, pyrochlore, and spinel, we first curate the reference set by downloading CIF files from Materials Project [10] that is likely to belong to each family based on formula and space group. We then use `fit_anonymous` method of the matcher to compare each generated structure to the structures in the reference set. A generated structure is considered matched if `fit_anonymous` returns true for at least one reference structure of the corresponding family. Note that this approach might result in false positive matches. For example, when we selected the reference set for pyrochlore, we downloaded CIF files Material Project that have composition A2B2O7. However, not all A2B2O7 are pyrochlore, so generated structures may still not be a pyrochlore despite being matched to one of the reference structures.

### A.2  Details of baselines

We use the following prompts in Table 7 to generate the CIF files for the end-to-end prompting baseline or to generate the chemical formulae for GenMS.

| Method | Prompt |
|---|---|
| Prompt CIF (baseline) | "I want you to generate another crystal information (CIF) files for a stable and potentially realistic material that belongs to {category}. [(Optional) Here are some information about {category} from Wikipedia.] Below are some examples of CIF files from this category: {example1, example2, ...} Please generate one more file for a crystal that is not in existing materials databases like Materials Project and ICSD. Please make sure the CIF file is valid. Just generate the file and do not say anything else." |
| Prompt formula (GenMS) | [(Optional) Here are some information about {category} from Wikipedia.] Please give me a list of chemical formulae for a hypothetical material for {category}. I want the formula to be stable, and potentially realistic and do not exist in dataset like Materials Project or ICSD. Please just give the formula and do not say anything else." |

Table 7: **LLM prompts for baseline and GenMS.**

### A.3  Compute, architecture, and training

We repurpose the 3D U-Net architecture [51, 52] into modeling atoms within a crystal structure by their $x, y, z$ locations concatenated with atom number (number of protons) $a$. As a result, we can

represent each crystal structure using an $Ax4$ matrix where $A$ is the total number of atoms in the structure, and the dimension with size $4$ represents the $x, y, z$ location and atom number of each atom. We repurpose the spatial downsampling and upsampling passes from typical U-Net for images or videos, and keep the resolution (number of points) the same, but still employ residual network with concatenating skip connections (see Figure 2 from the main text). Below we show the architecture and hyperparameters used in the diffusion model for crystals with compact representation.

| Hyperparameter | Value |
|---|---|
| Learning rate | 5e-5 |
| Optimizer | Adam ($\beta_1 = 0.9, \beta_2 = 0.99$) |
| Base hidden dimension | 256 |
| Hidden dimension multipliers | 1, 2, 4 |
| Number of mlp and self-attention blocks | 9 |
| Batch size | 512 |
| EMA | 0.9999 |
| Weight decay | 0.0 |
| Prediction target | $\epsilon$ |
| Attention head dimension | 64 |
| Dropout | 0.1 |
| Training hardware | 64 TPU-v4 chips |
| Diffusion noise schedule | cosine |
| Noise schedule log SNR range | [-20, 20] |
| Training steps | 200000 |
| Sampling timesteps | 256 |
| Sampling log-variance interpolation | $\gamma = 0.1$ |

Table 8: **Hyperparameters for training** the diffusion model in GenMS.

## A.4 Details of DFT calculations

In all our density functional theory (DFT) calculations, we employ the Vienna ab initio simulation package (VASP) [53, 54] with the Perdew-Burke-Ernzerhof (PBE) [55] functional and projector-augmented wave (PAW) potentials [56, 57]. Our computational settings align with those used in the Materials Project, as implemented in pymatgen [21] and atomate [58]. These settings include the application of the Hubbard U parameter to selected transition metals in DFT+U calculations, a plane-wave basis cutoff of 520 eV, specific magnetization settings, and the use of PBE pseudopotentials. However, we opt for updated versions of potentials for Li, Na, Mg, Ge, and Ga, maintaining the same valence electron count. For structural optimization, our protocol involves a two-stage relaxation of all geometric parameters, followed by a final static computation. We utilize the custodian package [21] to manage any issues with VASP and to make necessary adjustments to the simulations. Additionally, we generate gamma-centered k-points for hexagonal cells, deviating from the conventional Monkhorst-Pack scheme. We initialize our simulations with ferromagnetic spin, observing that attempts to explore alternative spin configurations were computationally too demanding. In our ab initio molecular dynamics (AIMD) simulations, we disable spin polarization and employ the NVT ensemble with a 2 fs timestep. For systems containing hydrogen, we reduce the timestep to 0.5 fs to ensure accuracy.

