# OpenReview forum: "Generative Hierarchical Materials Search"
_NeurIPS.cc/2024/Conference — NeurIPS 2024 poster_

### Official Review · Reviewer_DY2F · 2024-07-04

**Soundness:** 4
**Presentation:** 4
**Contribution:** 4
**Rating:** 7
**Confidence:** 3

**Summary:**

This paper proposes a hierarchical generative system for material search, which consists of a language model to translate user input into intermediate textual representation, a diffusion model which then generates crystal structure, and finally a property predictor used for sample selection.

The works leverage data at different levels from different sources: the high-level knowledge in web texts and low-level crystal information in specialized databases. This motivates the factorization of the text-to-crystal generation process into the high-level text-to-symbolic step, and the low-level symbolic-to-crystal step.

The generation results are filtered/ranked by a heuristic module for each step.

The approach is supported with experiments on several families of material.

**Strengths:**

The proposed framework effectively utilize data from multiple data sources.
The collaboration of the generation modules and scoring modules in the framework is a reasonable approach and might benefit other works. The discussion on future works are also useful.

**Weaknesses:**

The baselines seems to be weak. If the LLMs are finetuned on CIF data, probably the baseline performance can be improved.

**Questions:**

it seems the baseline is few-shot, but the GenMS language prompt is zero-shot.
Why? it is not easy to apply few-shot on GenMS or did you find it's not helpful compared to zero-shot?

**Limitations:**

the language model used in this paper, as well as potential ways to improve this module (such as prompting strategies, few-shot, finetuing) are not well explored.

---

> ### Author Rebuttal · Authors · 2024-08-07
>
> Thank you for the positive feedback of our work! Please find our response to your questions below.
>
> > LLM finetunine baseline
>
> The reason that we only included few-shot prompting baselines in the end-to-end evaluation of Table 1 is because, to our knowledge, there is no natural-language-to-crystal-structure dataset available for finetuning LLMs to generate crystals from loose natural language descriptions, as this work is among the first to study this problem. Hence, few-shot prompting is the most natural baseline. However, we do evaluate the subcomponents of GenMS to previous baselines, and show that the formula-conditioned generation in GenMS performs favourably compared to diffusion model [1] and finetuned language model baselines [2] in Table 5 and Table 6 of the paper.
>
> [1] Yang, M., Cho, K., Merchant, A., Abbeel, P., Schuurmans, D., Mordatch, I., & Cubuk, E. D. (2023). Scalable diffusion for materials generation. arXiv preprint arXiv:2311.09235.
>
> [2] Antunes, Luis M., Keith T. Butler, and Ricardo Grau-Crespo. "Crystal structure generation with autoregressive large language modeling." arXiv preprint arXiv:2307.04340 (2023).
>
> > Why GenMS uses zero-shot prompting but baselines use few-shot prompting
>
> We consider GenMS in a zero-shot prompting setting because of the practical motivation of materials discovery, i.e., structures to be discovered may not have similar existing structures to use as few-shot prompt. However, when we tried zero-shot prompting of cif files using LLMs directly, we observed close to zero percent valid cif files being generated. Therefore we had to include a few cif file examples as input to the LLMs to specify the format of the cif files.
>
> > Details of the LLM used and potential ways to improve the LLM module
>
> Per the reviewer’s suggestion, in addition to Gemini long context, we have also used GPT4 as the LLM module, and observe that GPT4 performs similarly (and occasionally better) than Gemini long context. We did not further tune the prompts, which may lead to different results.
>
> | Metrics | GEMS-Gemini | GEMS-GPT4 |
> | ----------- | ----------- | ----------- |
> | CIF validity | 1 | 1 |
> | Structural validity | 1 | 1 |
> | Composition validity | 0.85 | 0.97 |
> | Formation energy (GNN) | -0.47 | -0.95 |
> | Formation energy (DFT) | -1.32 | -1.39 |
> | Uniqueness | 0.9 | 0.9 |
> | Match rate | 0.93 | 0.95 |
>
> We have also conducted two additional studies for RAG. First, we eliminate half of the retrieved Wiki text, and observe the match rate in Table 4 drops from 0.89 to 0.85. Second, we replace the text retrieved from Wikipedia by text retrieved from Materials Project [1] (entries that contain the keyword Perovskite), and observe that the match rate in Table 4 drops to 0.86, which is still better than no retrieval at all. We will include these additional results in the final version of the paper. We think there are many other interesting aspects of the system that can be improved, such as using research articles to further enhance RAG. We see our work as the first that calls for further contribution for language-to-structure generation.

---

> > ### Comment · Reviewer_DY2F · 2024-08-12
> >
> > Thanks for your response! My concerns/questions have been addressed.

---

### Official Review · Reviewer_SSy6 · 2024-07-09

**Soundness:** 3
**Presentation:** 2
**Contribution:** 2
**Rating:** 5
**Confidence:** 3

**Summary:**

This work leverages the powerful capabilities of large language models to propose a generative framework that transitions from natural language to material structures. The framework divides material generation into high-level semantic information, which can be expressed as chemical formulas in text, and low-level structural information, which requires atomic types and coordinates for expression. High-level semantic information is generated by the large language model based on user instructions, while the low-level structural information is completed by a conditional diffusion model. Both high-level and low-level information are optimized and ranked using empirical formulas. Finally, a property-predicting Graph Neural Network (GNN) is used to further optimize the generated material structures. The proposed model accomplishes end-to-end material generation, particularly starting from user instructions, which makes this research highly practical.

**Strengths:**

1. Leveraging the capabilities of large language models to generate materials based on user instructions is a highly imaginative approach, with significant potential for future applications.
2. The overall structural design is comprehensive, including various component models and empirical functions.
3. There are technical innovations, such as the use of compact crystal representations to accelerate inference.

**Weaknesses:**

1. The claim of "end-to-end" is confusing. The overall model seems more like a combination of different components (LLM, diffusion model, GNN). These components are not inherently connected and are not trained together, which seems to create a gap with the typical "end-to-end" meaning in the machine learning field.
2. The work seems more like an engineering contribution. While I do not deny the significance of such work in the current LLM era, the extensive discussion on multi-objective optimization (Section 2) feels somewhat redundant. This part appears to involve simple rule-based pruning, making the emphasis on multi-objective optimization as a contribution rather unconvincing.
3. The work lacks ablation studies. For instance, as mentioned in point 2, many rules are designed based on heuristic experience (including how to filter chemical formulas and molecular structures). I am uncertain about the actual impact of these heuristic rules and their robustness.
4. The core comparison baseline in this paper is the ability of a generic LLM without fine-tuning to generate materials. This is evidently unfair when compared to the proposed complete framework, which includes the diffusion model. Even adding fine-tuned LLMs does not seem convincing. The authors should at least compare their work with some specialized material generation models based on diffusion models or other methods.
5. Similar to point 1, although it claims to be end-to-end, it seems that only the diffusion model is trained. I believe there is a lack of detailed explanations regarding the LLM and the property-predicting GNN used.

**Questions:**

1. Lacking domain knowledge, I am curious about the guiding capability of the chemical formula in this context. Why is it that a chemical formula generated by an LLM based on user instructions is considered so meaningful in providing guidance?
2. Using compact crystal representations to accelerate inference is understandable, but I am still curious why the authors chose a U-Net architecture as the denoising model. Typically, in non-image domains, common backbones are GNNs or Transformers.

**Limitations:**

1. While generating molecules through user natural language instructions is a very exciting prospect, it seems that the role of the language model in this context is merely to generate a chemical formula. This makes it feel like the potential of the LLM is quite limited.

---

> ### Author Rebuttal · Authors · 2024-08-07
>
> Thank you for the detailed review. Please see our response below.
>
> > The "end-to-end" terminology.
>
> We agree that our usage of "end-to-end" is not conventional and can be misleading. "End-to-end" in this paper’s context refers to end-to-end automation from formulae proposal to structure generation and verification, as opposed to end-to-end modeling. We have clarified the usage of "end-to-end" in the paper and reduced its frequency for clarity.
>
> > The multi-objective optimization formulation
>
> Thank you for recognizing this work’s contribution despite its simplicity. We believe that formally formulating the materials search problem from natural language input is important, so that many problems in materials discovery can be considered under the same framework. Under the formulation of the multi-objective optimization, rule-based pruning is only one proposed approach to solve this objective. One can also perform constrained optimization, planning, or reinforcement learning with external feedback to solve this multi-objective optimization problem, which we consider to be interesting future work.
>
> > Heuristic rules and their effectiveness
>
> We would like to clarify that the heuristics considered in this paper are only examples of many constraints and heuristics that practitioners in materials science may want to enforce, as opposed to a set of hand-designed heuristics that work the best for materials discovery. The current heuristics, especially those in $R_\text{hi}$, are only basic checks on instruction following. As practitioners use this framework, we anticipate additional constraints and heuristics to be enforced depending on the specific types of materials being discovered and the additional domain knowledge of the respective users. We have rewritten line 116-120 to highlight that $R_\text{hi}$ considered in this paper are only examples of high-level heuristics that can be optimized.
>
> > Comparison to finetuned LLM and diffusion models.
>
> The reason that we only included few-shot prompting baselines in the end-to-end evaluation of Table 1 is because, to our knowledge, there is no natural-language-to-crystal-structure dataset available for finetuning LLMs to generate crystals from loose natural language descriptions, as this work is among the first to study this problem. Hence, few-shot prompting is the most natural baseline. However, we do evaluate the subcomponents of GenMS to previous baselines, and show that the formula-conditioned generation in GenMS performs favourably compared to diffusion model [1] and finetuned language model baselines [2] in Table 5 and Table 6 of the paper. [1] and additional work on diffusion models has demonstrated better performance than GNNs ([3]), hence we did not include comparison to GNN generation.
>
> [1] Yang, et al. Scalable diffusion for materials generation. arXiv preprint arXiv:2311.09235 (2023).
>
> [2] Antunes, et al. Crystal structure generation with autoregressive large language modeling. arXiv preprint arXiv:2307.04340 (2023).
>
> [3] Zeni, Claudio, et al. "Mattergen: a generative model for inorganic materials design." arXiv preprint arXiv:2312.03687 (2023).
>
> > Detailed explanation of LLM and property GNN used
>
> The LLM and GNN used are described in detail in the previous work [1] and [2]. We have included these detailed descriptions in the appendix for the final version of this paper.
>
> [1] Reid et al. Gemini 1.5: Unlocking multimodal understanding across millions of tokens of context. arXiv preprint arXiv:2403.05530, 2024
>
> [2] Amil Merchant, Simon Batzner, Samuel S Schoenholz, Muratahan Aykol, Gowoon Cheon, and Ekin Dogus Cubuk. Scaling deep learning for materials discovery. Nature, 624(7990):80–85, 363 2023.
>
> > Why chemical formula
>
> Chemical formulas are both connected to low-level properties of structures (e.g., conductivity) and high-level semantics that a user might specify using natural language (i.e., a crystal with carbon and silicon). Hence, chemical formulas are natural intermediate information. Moreover, chemical formula exists both in high-level datasets such as textbooks, wiki articles, and research papers, as well as in low-level crystal structure databases such as Materials Project [1] and ICSD [2], making them a natural interface as LLMs and diffusion models are trained on high and low level datasets, respectively.
>
> [1] Jain, Anubhav, et al. "Commentary: The Materials Project: A materials genome approach to accelerating materials innovation." APL materials 1.1 (2013).
>
> [2] Hellenbrandt, Mariette. "The inorganic crystal structure database (ICSD)—present and future." Crystallography Reviews 10.1 (2004): 17-22.
>
> > Why UNet
>
> We have experimented with different architectures including only using transformers with local residual connections as opposed to a UNet-like architecture, but these alternatives did not work as well in practice. We did not experiment with GNNs for generation, because GNNs with graphs represented by adjacency matrices scale quadratically in the number of atoms. On the other hand, the modeling complexity of GenMS only scales linearly with the number of atoms, as the compact crystal representation of GenMS does not consider edges.
>
> > Limitations of only generating chemical formula
>
> Generating formula is in fact nontrivial, especially for complex structures such as metal-organic frameworks (MOFs), which often have long formulae. Formulae implicitly capture rich human knowledge such as charge balancing, conductivity, etc. This knowledge is implicitly captured by LLMs and elicited during the formula generation process. That being said, we believe that formulae do not have to be a hard interface between high-level semantic and low-level structure generation. Exploring alternatives like including additional textual information such as space groups are interesting future works to explore, as well as directly generating structures from loose natural language, once such a dataset becomes available for finetuning LLMs.

---

> > ### Comment · Reviewer_SSy6 · 2024-08-10
> >
> > Thanks for the rebuttal. I appreciate the authors' efforts in addressing the concerns raised. However, I still have some reservations about this paper. First and foremost, I would like to acknowledge the contribution this work makes to the field of text-to-material generation. However, given that the core of this paper is about demonstrating a comprehensive system for generating materials from text, I believe more emphasis should have been placed on showcasing the potential of this process.
> >
> > In particular, the paper seems to lack sufficient experimental evidence that demonstrates the alignment between text prompts and the generated materials, especially with regard to specific properties or characteristics. The only relevant experiment I observed was the one regarding limestone, which primarily demonstrated that the generated materials contained a specific element and metal as specified by the text input.
> >
> > I am not an expert in the field, so my perspective might be limited, but I am struggling to understand the advantage of using text prompts to control such low-level information, as opposed to having domain experts directly design chemical formulas that can then be used by a diffusion model to generate materials.
> >
> > In conclusion, I believe that this paper does not make a particularly strong contribution in terms of technical novelty, and it also falls short in showcasing the true potential of text-to-material generation (as most experiments seem to focus on the reasonableness and novelty of the generated materials, with little emphasis on how well they align with the original text prompts).

---

> > > ### Author Response · Authors · 2024-08-12
> > > **Reply to Reviewer SSy6**
> > >
> > > Thank you for engaging in discussion! First, we would like to highlight that we have indeed focused on evaluating the alignment between text prompt and generated materials in Table 1, Table 3, Table 4, and Figure 3, including evaluating alignment of properties and characteristics. Specifically, the “Match rate” metric in Table 1 measures the geometric alignment between generated structures and known structures from the family specified by the language input. In Table 3, we showcase that language prompts can flexibly control elements contained and conductivity, as well as being used to encourage uniqueness. In Table 4, the “Match rate” again measures alignment of geometric properties between language prompt and generated structures. In Figure 3, we showcased the alignment of language and material while considering prompts that imply chemical elements (Elpasolite implies sodium potassium aluminum fluoride) and geometric properties (double perovskite and layered material). We would like to emphasize that measurements of geometric alignment are highly reflective of property alignment, as chemical properties are often determined by geometry of crystals structures.
> > >
> > > Second, while chemical properties might seem too low-level to be controlled by language, there are often high-level patterns of properties that can benefit from language control. For instance, conductivity is determined by band gap (a low-level property), but metallic elements are more likely to lead to conductive materials, while semimetals (e.g., boron, silicon, germanium) may conduct depending on what other elements are in the structure. Such high-level knowledge often exists in webpages and research papers, and language models can utilize these knowledge to suggest appropriate formulas.
> > >
> > > Third, having domain experts directly design chemical formulas can be difficult, because certain properties and characteristics such as formation energy are often determined by low-level structures, which is exactly where GenMS shines in its ability to integrate high and low level information.

---

> > > > ### Comment · Reviewer_SSy6 · 2024-08-12
> > > >
> > > > Thanks for the response. After carefully reviewing the detailed rebuttal provided by the authors, I have decided to raise my score to a borderline accept. I acknowledge that this paper may indeed represent the first attempt to accomplish a text-to-material generation task.
> > > >
> > > > The experimental results, while showing potential, demonstrate the ability to capture higher-level semantics through text and subsequently generate materials that meet certain abstract properties at a lower level.
> > > >
> > > > However, I must clarify that I lack sufficient domain knowledge in this area, which limits my ability to objectively assess the solidity of the experiments. Additionally, I perceive this work as more of an engineering effort that has constructed a preliminary but promising integrated system, rather than offering significant technical novelty. As a result, I am lowering my confidence score by 1.

---

### Official Review · Reviewer_Tzq3 · 2024-07-12

**Soundness:** 3
**Presentation:** 3
**Contribution:** 3
**Rating:** 5
**Confidence:** 1

**Summary:**

The paper presents a language-to-structure generation model for crystal structures. They propose a hierarchical approach that uses a language model to generate intermediate textual crystal information and then generate low-level crystal structures using diffusion models. They demonstrate their ability to generate crystal structures from language instructions and compare it to baselines. Results show that GenMS outperforms the baselines in generating valid and low-energy structures that satisfy user requests.

**Strengths:**

1. The paper proposes a technically sound approach that decomposes the problem into two step and uses separate models for each step based on the task characteristics.
2. The empirical results are good. They demonstrate their effectiveness in generating valid and low-energy structures satisfying user requests and they outperform baselines.

**Weaknesses:**

1. The baseline is relatively weak as it is only a few-shot prompted LLM. For example, the ablation of each of their proposed modules (e.g., LLMs, retrieval) should be treated as baselines and the results should be put in the main table with more comprehensive evaluation results than the current ones.
2. The overall workflow is a combination of several widely-used models (e.g., LLMs, diffusion models, retrieval models). The pipeline makes sense but combining these models may not be enough to make significant contributions for a research paper.

**Questions:**

Can you add stronger baselines and show more comprehensive evaluation results?

**Limitations:**

the authors adequately addressed the limitations

---

> ### Author Rebuttal · Authors · 2024-08-07
>
> Thank you for the thoughtful feedback. Please see our response below.
>
> > More comprehensive baselines
>
> The reason that we only included few-shot prompting baselines in the end-to-end evaluation of Table 1 is because, to our knowledge, there is no natural-language-to-crystal-structure dataset available for finetuning LLMs to generate crystals from loose natural language descriptions, as this work is among the first to study this problem. Hence, few-shot prompting is the most natural baseline. However, we do evaluate the subcomponents of GenMS to previous baselines, and show that the formula-conditioned generation in GenMS performs favourably compared to diffusion model [1] and finetuned language model baselines [2] in Table 5 and Table 6 of the paper.
>
> [1] Yang, M., Cho, K., Merchant, A., Abbeel, P., Schuurmans, D., Mordatch, I., & Cubuk, E. D. (2023). Scalable diffusion for materials generation. arXiv preprint arXiv:2311.09235.
>
> [2] Antunes, Luis M., Keith T. Butler, and Ricardo Grau-Crespo. "Crystal structure generation with autoregressive large language modeling." arXiv preprint arXiv:2307.04340 (2023).
>
> > Concerns around contribution for a research paper.
>
> We would like to highlight the nontrivial aspects of this work. Even though the solution of combining LLMs with diffusion models might seem trivial, the reason behind such a design and the process of how GenMS is the first to arrive at this design is nontrivial. Our contributions include recognizing the knowledge gap in high-level textbooks and low-level materials structure datasets, recognizing the gaps between models of different abstraction levels (LLMs for high-level semantics and diffusion models for low-level details), and properly addressing these gaps using both models at the abstraction level with abundant data. We hope that these insights, as well as our novel hierarchical, multimodal search algorithm may inspire solutions to problems in other domains that involves discovery using natural language.

---

> > ### Author Response · Authors · 2024-08-12
> > **Author follow-up**
> >
> > Dear Reviewer,
> >
> > We would like to ask if your questions and concerns have been addressed by our response, and if there is anything else preventing you from increasing your score. Please let us know, and thank you for your time.

---

> > ### Comment · Reviewer_Tzq3 · 2024-08-12
> > **Thank you for the response**
> >
> > Thank you for the response!
> >
> > I've read all the responses and the other reviews. The response does not address my concern about the technical contributions of the paper, as the overall workflow still seems like a combination of several widely-used models. However, because I am not an expert in this field and this work may be interesting to researchers in this particular domain, I would keep my relatively positive rating as is.

---

### Official Review · Reviewer_Xu7Y · 2024-07-15

**Soundness:** 3
**Presentation:** 3
**Contribution:** 3
**Rating:** 6
**Confidence:** 2

**Summary:**

This paper presents Generative Hierarchical Materials Search (GenMS), a novel framework for material search using LLMs. GenMS consists of a language model that takes high-level natural language as input and generates intermediate textual information about crystals (e.g., chemical formulae). Specifically, it uses a RAG architecture to make an LLM incorporate information from Wikipedia articles and other retrieved articles from the Internet. The generated formulae are filtered and re-ranked by a heuristic function involving domain knowledge, such as a formulae regular expression checker. A diffusion model then generates material structures conditioned on the filtered formulae. Experimental results show that GenMS is a solid framework that can generate structures satisfying users' requirements, demonstrating that using LLMs for structure search is a promising direction.

**Strengths:**

1. The paper is easy to follow, even for readers not familiar with materials search.
2. The proposed method is interesting, particularly in its use of RAG  to enable LLMs to follow the formula in articles. The ablation study shows RAG's effectiveness.
3. Using LLMs for material search or other scientific problems is an important but underexplored domain. This paper presents a solid framework for using LLMs in material structure search, showing that this is a promising direction for scientific applications.

**Weaknesses:**

1. The architecture involves several steps requiring ad-hoc decisions, such as using Wiki-API for information retrieval or heuristic operations for selecting formulae. For example, lines 117-120 describe ad-hoc operations to prune intermediate formulae, making the method hard to reproduce and requiring more human effort. There might be more elegant, machine-learning-based ways to incorporate these operations, such as using LLMs for formula checking.
2. The heuristic function design involves significant domain knowledge, requiring more ablation studies or analysis to justify the heuristic choices.

**Questions:**

1. Considering the complexity of the whole system, what is the average inference latency for each query?
2. How does using Wiki-API affect performance? Is there any ablation study on combining Wiki-API and RAG differently?

**Limitations:**

The design of the heuristic function requires a lot of domain knowledge, making the method difficult to reproduce.

---

> ### Author Rebuttal · Authors · 2024-08-07
>
> Thank you for recognizing the significance of this work! Please see our response to your questions below.
>
> > Ad-hoc decisions in the framework.
>
> We would like to clarify that the choice of $R_\text{hi}$ and $R_\text{lo}$ are not ad-hoc decisions made by the framework, but highlighting what practitioners can enforce in the materials search process. Checks currently implemented in the current framework are rather basic, e.g., formulae containing the desired elements, formulae consist of valid chemical elements, don’t contain duplicates, and have desired element ratios. There are certainly other checks that can be incorporated into the current framework, such as "must contain element Si" and "have less than 5 chemical elements", which gives a domain expert additional control in the generation process. We have rewritten lines 116-120 to reflect that $R_\text{hi}$ and $R_\text{lo}$ are only examples of what can be optimized in GenMS, as opposed to arbitrary choices we made.
>
> > Additional ablations.
>
> As discussed in the previous paragraph, the goal of GenMS is not to develop the best heuristics but to allow diverse checks and constraints to be incorporated. That being said, the choice of what information to include in RAG is indeed an interesting ablation. We conducted two additional studies. First, we eliminate half of the retrieved Wiki text, and observe the match rate in Table 4 drops from 0.89 to 0.85. Second, we replace the text retrieved from Wikipedia by text retrieved from Materials Project [1] (entries that contain the keyword Perovskite), and observe that the match rate in Table 4 drops to 0.86, which is still better than no retrieval at all. We will include these additional results in the final version of the paper. However, we were unable to conduct more ablations, such as using research articles containing keywords as retrieval text in RAG due to the lack of a dataset with easily retrievable research articles in materials science. We leave this investigation for future work.
>
> [1] Jain, Anubhav, et al. "Commentary: The Materials Project: A materials genome approach to accelerating materials innovation." APL materials 1.1 (2013).
>
> > Incorporating LLMs for formula checking.
>
> We agree that using LLMs for formula checking is more general than using regular expression checkers. In fact, we are already using LLMs to rank the top W formulas as described on line 122. To investigate the reviewer’s suggestion of LLM feedback, we use Gemini long context [1] to judge whether structures are nonmetals. For 25 known nonmetals, LLM determins 23 are nonmetals, achieving a success rate of 92%. However, using LLM feedback for *novel* structures is more challenging, as we do not have ground truth for whether the LLM feedback is correct.
>
> We believe that similar to RL with feedback for LLMs, where diverse heuristic and reward functions such as human feedback, AI feedback, and machine feedback have been jointly used to improve system performance, GenMS can also benefit from diverse heuristics and sources of feedback, including feedback from LLMs.
>
> [1] Machel Reid, Nikolay Savinov, Denis Teplyashin, Dmitry Lepikhin, Timothy Lillicrap, Jean365 baptiste Alayrac, Radu Soricut, Angeliki Lazaridou, Orhan Firat, Julian Schrittwieser, et al.
> 366 Gemini 1.5: Unlocking multimodal understanding across millions of tokens of context. arXiv
> 367 preprint arXiv:2403.05530, 2024
>
> > Latency of the system
>
> Thank you for bringing up this question. We have measured the latency of the system on TPU-V4 with batching enabled (using batch size 64 for the structure branch and energy GNN), and report the results below.
>
> | Language Branch (H): 1 | Structure Branch (L): 1 (batch size 64) | Energy GNN (batch size 64) | Total |
> | ------------ | ----------- | ----------- | ----------- |
> | 43ms  | 100ms | 800 ms | 943ms |
>
> | Language Branch (H): 25 | Structure Branch (L): 100 | Energy GNN (batch size 64) | Total |
> | ------------ | ----------- | ----------- | ----------- |
> | 1150ms | 4600ms | 36690ms | 42440 ms |
>
> The latency of the language branch is measured on Gemini long context model and latency of structure branch is measured on the compact diffusion model. The latency of the energy prediction GNN is high due to the paddings applied to the structures to pad structures to the same number of nodes, and due to the breadth of the search (i.e. predicting energies for 2500 structures). We will include this result in the final version of the paper.
>
> > Concerns around the requirement of domain knowledge
>
> While it is compelling to eliminate the need for any domain knowledge, we think that some level of domain knowledge is hard to avoid, especially given the end goal is to have materials scientists use the framework to discover and even synthesize novel materials. Therefore, the framework should provide a way for material scientists to specify and verify the generation. Instead of getting rid of domain knowledge, GenMS leverages domain knowledge both through in-context learning and through heuristic/reward design, which gives practitioners the flexibility to generate and verify crystals based on their need.
>
> That being said, we are not leveraging more domain knowledge to evaluate GenMS than previous work such as [1] and [2]; the validity metrics in Table 1 are standard in previous generative models for materials work. The results of GenMS are easy to reproduce, with the language prompt provided in Appendix A.2 and validity metrics following previous work ([1], [2]).
>
> [1] Xie, Tian, et al. "Crystal diffusion variational autoencoder for periodic material generation." arXiv preprint arXiv:2110.06197 (2021).
>
> [2] Yang, M., Cho, K., Merchant, A., Abbeel, P., Schuurmans, D., Mordatch, I., & Cubuk, E. D. (2023). Scalable diffusion for materials generation. arXiv preprint arXiv:2311.09235.

---

> > ### Author Response · Authors · 2024-08-12
> > **Author follow-up**
> >
> > Dear Reviewer,
> >
> > We would like to ask if your questions and concerns have been addressed by our response. Please let us know, and thank you for your time.

---

### Decision · Program_Chairs · 2024-09-25

**Decision:**

Accept (poster)

**Comment:**

This paper introduced GenMS, a framework that uses LLMs to generate novel material structures based on high-level NL input using chemical formulae. For the inference they combine LLMs with a diffusion model and domain-specific heuristics. They showcase promising results in generating structures that satisfy user requirements. This is an interesting new direction towards materials discovery and design.

The paper includes ad-hoc decisions in their framework, making the method difficult to reproduce. Authors clarified that the ad-hoc decisions are not arbitrary, but rather examples of what practitioners can enforce in the materials search process. The authors have also conducted additional experiments and demonstrated the effectiveness of the RAG architecture and the impact of using different retrieval texts. The authors acknowledge that some level of domain knowledge is unavoidable, but it would be beneficial to provide more guidance on how to incorporate domain knowledge into the framework.

Another point that was raised is that the authors should include stronger baselines and more comprehensive evaluation results. The authors explained that the few-shot prompting baseline is the most natural choice due to the lack of a natural-language-to-crystal-structure dataset, but also they pointed to additional evaluations in the paper. The reviewer has also raised the issue for the paper that it is a combination of widely-used models, questioning whether this is enough to make significant contributions for a research paper. The authors responded by highlighting that the approach recognizes knowledge gaps and properly addresses them using a novel hierarchical, multimodal search algorithm.

The third reviewer has pointed to the paper's technical novelty and the lack of experimental evidence. The authors have provided additional evidence but the reviewer still thinks that the alignment between text prompts and generated materials still remains particularly with regard to specific properties of characteristics.

The contributions of this paper is significant, as they pave the way for further progress in the materials discovery problem. Although the evaluations and results may not be on par with other machine learning papers in the natural sciences domain (e.g., biology), this is a novel problem to tackle with ML.

I suggest the following minor revisions to improve the quality of the paper (which can be done easily):
-- provide more guidance on how to incorporate domain knowledge into the framework.
-- the authors provided latency measurements, but it would be good to discuss potential strategies for reducing latency.